# The Frequency of Errors in Determining Age Based on Selected Features of the Incisors of Icelandic Horses

**DOI:** 10.3390/ani9060298

**Published:** 2019-05-30

**Authors:** Jarosław Łuszczyński, Magdalena Pieszka, Weronika Petrych, Monika Stefaniuk-Szmukier

**Affiliations:** Department of Horse Breeding, Institute of Animal Science, Agricultural University, Al. Mickiewicza 24/28, 30-059 Cracow, Poland; jaroslaw.luszczynski@urk.edu.pl (J.Ł.); punktur.weronika@gmail.com (W.P.); monika.stefaniuk-szmukier@urk.edu.pl (M.S.-S.)

**Keywords:** Icelandic horses, equine incisors, ageing horses

## Abstract

**Simple Summary:**

Methods of assessing the age of horses based on the appearance of teeth, although used for many years, seem to suffer from relatively large errors. However, this method can play an auxiliary role in identifying horses of unknown origin. Furthermore, this method is a useful tool for owners, breeders, or veterinary surgeons to make decisions regarding purchase, insurance, or treatment and provides information on the specific characteristics of horses of different breeds. This study aimed to assess the suitability of selected features of the incisors for the determination of the age of Icelandic horses. Determining the age of Icelandic horses based on the appearance of teeth only matched the real age in more than one-third of the 126 individuals assessed. The average percentage of errors made in the assessment of age based on the eruption and growth of deciduous incisors was significantly smaller compared to determining age based on replacement of deciduous to permanent incisors, the disappearance of cups, and/or changes in the shape of the grinding surfaces. It is likely that characteristic changes occurring in Icelandic horses’ incisors may be related to the specific course of development processes of this breed.

**Abstract:**

The structure and changes occurring to horses’ teeth during ontogeny are not only used to assess the degree of somatic maturity but also the development of universal patterns and is therefore used to determine the age of horses. Research shows that methods of assessing the age of horses based on the appearance of teeth tend to suffer from relatively large errors. This is probably influenced by the results of intensive selection and being kept in living conditions that differ substantially from their natural environment. This study aimed to assess the suitability of selected features of the incisors to determine the age of Icelandic horses. One hundred and twenty-six Icelandic horses (78 mares and 48 stallions) of different ages (range: 0–24 years; groups: 0–2 years, >2–5 years, >5–11 years, and >11 years) were examined by an experienced horse person who was blinded to the actual age of the horse and did not know which age group horses were in. Age was determined by the inspection of each horse’s teeth and was compared to the actual age of the horse recorded in the breeding documentation, and the percentage of mistakes made regarding the age group was calculated. The estimated age did not match the real age in 36.5% of the horses. The age was more often underestimated (19.0%) by, on average, 0.9 ± 1.0 years than overestimated (17.5%) by, on average, 1.3 ± 1.4 years. Within age groups, the least number of errors in determining age were made in young horses aged 0–2 years, when the eruption and growing of the deciduous incisors and the disappearance of the cups was considered. The average percentage of errors in this group (2.1%) was significantly lower (*p* < 0.01) than for older horses, whose age was estimated based on the exchange of deciduous to permanent teeth (55.8%), disappearance of the cups (68.0%), and shape changes on the grinding surface (40.0%). Significantly more frequent underestimation of age based on replacing deciduous for permanent incisors and significantly more frequent overestimation of age on the basis of the disappearance of the cup may indicate that Icelandic horses up to 5 years of age are characterized by a slower rate of growth than horses of other breeds, especially warmblood horses. These results suggest that patterns used to determine the real age of horses based on changes occurring on the incisors should be modified in order to consider the specificity of the course of growth and maturation processes of horses of various types and breeds.

## 1. Introduction

The origin of the Icelandic horse dates back to the 9th century and is associated with the arrival of Norwegian settlers to Iceland who brought horses of pony type with them [1,2]. However, analysis based on the mtDNA and microsatellite data indicates that Icelandic horses are more closely related to Mongolian than Norwegian horses [3]. Bjørnstad and Røed [4] also suggest that the Icelandic horse population may originate from Shetland and Nordland breeds as well as the Fjord Pony and Coldblooded Trotter. The ban on importing animals to Iceland, introduced in the 10th century to prevent the spread of diseases, created a consolidated, uniform breed and protected Icelandic horses against crossbreeding with other breeds, especially with Oriental horses imported into the European continent [5]. Icelandic horses, due to their specific anatomical and physiological features, easy-going nature, and ability to use alternative gaits, are becoming ever more popular around the world. More than 50% of the 200,000+ Icelandic horses around the world are kept in other countries besides Iceland [6]. Their prolonged isolation, purity of breeding, and maintenance in a stable-free system close to the wild environment has led the Icelandic horse to become a valuable research model of various morphological traits, including teeth. During evolution, to adapt to the changing climatic and natural conditions, equine teeth transformed and developed characteristic features such as delayed termination of root formation, prolonged dental growth and eruption—the horse teeth belong to the hypsodont type, according to the shape of the grinding surface, disappearance of cups, occurrence of dental stars, appearance and position of the Galvayne’s groove, and occurrence of upper corner incisor hooks [7,8,9]. Eruption and replacement of the deciduous teeth with permanent teeth and the disappearance of the records occurring on the horse’s incisors during ontogenesis occur regularly and are not only used to assess the degree of somatic maturity [10] but also the development of universal patterns, which allows one to determine the age of horses (Table 1). Research results show that methods of assessing the age of horses based on the appearance of teeth, although used for many years, seem to suffer from relatively large errors [9]. This is probably influenced by intensive selection and living conditions, especially the nutrition of modern horses based on complete mixtures, concentrates, and special additions, which differs significantly from their natural environment where nutrition of horses is based on the grass. However, despite doubts about the effectiveness of this method, it can play an auxiliary role in identifying horses of unknown origin [11]. Furthermore, it is a useful tool for owners, breeders, or veterinary surgeons to make decisions regarding purchase, insurance, or treatment and provides information on the specific characteristics of horses of different breeds. This study aimed to assess the suitability of selected features of the incisors to determine the age of Icelandic horses.

## 2. Material and Methods

One hundred and twenty-six Icelandic horses (78 mares and 48 stallions) of different ages (from birth to 24 years) from a stud farm in southwestern Poland were examined in 2018. The stud farm used a grazing system for raising and keeping horses. During the summer season, the horses were turned out and had free access to pasture, grazing, and water. In winter, once a day for about two hours, horses in pasture groups were put in the wooden free-range stable bed with straw and fed with free access to meadow hay served from a floor. The period in the stable was used to carry out the planned experiment. The assessment was performed by an experienced horse person with extensive experience in determining age based on the appearance of the teeth but no specialization in veterinary equine dentistry. The examination included extending the tongue into the diastema area to prevent closure of mouth. The visible parts of the upper and lower jaws were visually inspected for the eruption of deciduous incisors, the replacement of permanent incisors, the disappearance of the cups, and any change of shape on the grinding surface. Using these parameters, the age was determined according to the method included in the instruction for written and graphical descriptions of horses and donkeys [12] (Table 1). The identity of each of the examined horses was confirmed using microchip data. The horses were divided into age groups by assigning a specific characteristic of incisors’ teeth to each of them to enable age recognition (Table 2). The age was determined based on the inspection of the teeth and was compared to the actual age of the horse recorded in the equine passport, and then the percentage of mistakes made by age group was calculated. The chi-square goodness of fit test identified if there were significant differences in the percentage distribution of errors. Analysis was undertaken using Statistica for Windows 13.0 (TIBCO Software Inc., Palo Alto, CA, USA).

## 3. Results

An analysis of the appearance of the dentition on the studied population, indicated that the age of Icelandic horses determined on this basis did not agree with the real age in 36.5% of the horses (Table 2 and Table 3). The age was more often underestimated by, on average, 0.9 ± 1.0 years than overestimated (17.5%) by, on average, 1.3 ± 1.4 years.

Within age groups, the least number of errors in determining age based on the appearance of teeth occurred in young horses aged 0–2 years, when the eruption and growing of the deciduous incisors and the disappearance of the cups was considered. The average percentage of errors in this group (2.1%) was significantly lower (*p* < 0.01) than for older horses, whose age was estimated based on the exchange of deciduous to permanent teeth (55.8%), disappearance of the cups (68.0%), and shape changes on the grinding surface (40.0%). In horses aged 0–2 years, in only one case the age of the horse was reduced by one year due to the delayed eruption of the deciduous corner incisors. In the >2–5 years age group, the age of horses was significantly more often (*p* < 0.05) underestimated (41.9% of 43 horses in the group) than overestimated (13.9%). Errors related to underestimation of age were due to a delayed exchange of deciduous incisors for permanent incisors (in eight horses—central incisors, in six horses—intermediate incisors, and in three horses—corner incisors). The most common reason for overestimating the age was the accelerated exchange of intermediate incisors. There were also observed instances of earlier exchange of central and corner incisors from deciduous to permanent and the quicker disappearance of cups on the lower central incisors. All these deviations from the standard approach resulted in the age of Icelandic horses in this group to be underestimated by an average of 0.5 years or overestimated by, on average, 0.8 years. In contrast to the previous group, the ages determined based on the appearance of teeth in Icelandic horses in the range of >5–11 years were significantly (*p* < 0.05) more often underestimated (56.0%) than overestimated (12.0%). Errors related to overestimating age were associated with a faster abrasion of the permanent incisors and thus an earlier disappearance of the cups. Most often, there were no cups on the lower central incisors of 5-year-old horses, whereas in other horses, the cups on the lower intermediate incisors, as well as on the upper central and corner incisors, were disappearing from 1 to 3 years earlier. The lowering of the age was caused by a slower abrasion of the lower and upper intermediate incisors and the associated later disappearance of cups on these teeth. The indicated abnormalities caused the average difference between the real age and the estimated age based on the appearance of dentition to be both overestimated and underestimated by 1 year. In analyzing the errors made when estimating the age of Icelandic horses older than 11 years, half of the error cases in this age group were overestimated and half of the error cases were underestimated. Overestimating age by an average of 5 years was associated with an earlier change of shape on the grinding surface; in one case, the surface of upper corner incisors changed from transversely oval to round, and in the second the surface of upper intermediate incisors changed from round to triangular. Lowering the age by an average of 3 years was due to the delay in changing the shape of the grinding surface of the upper central incisors from transversely oval to round or lower intermediate incisors from round to triangular.

## 4. Discussion

The results of this work and previous studies by other authors [10,11,13,14,15,16,17,18,19,20,21] show that the actual age of horses cannot always be reliably determined based on specific features of the incisors. In the case of Icelandic horses, it has been shown that more than one-third of the 126 tested individuals had a mistaken estimated age when using this method. A similar percentage of errors (35%) in the determination of age was recorded in Hucul horses (*n* = 173) [21], while in Arabian (*n* = 97), Anglo-Arabian (*n* = 55), and polish warmblood horses (*n* = 133), this error rate was much higher and accounted for 73%, 71%, and 69% of the horses, respectively [10,22]. Twice the difference in mistakes determining the age based on the appearance of the teeth between primitive and warmblood horses may suggest the existence of a significant type or breed effect on the course of changes occurring on horse incisors during ontogenesis. This dependence was also indicated by Gáspárdy et al. [20] in studies carried out on horses of various breeds bred in Hungary and Germany and by Muylle et al. [16,17] in draft horses, trotters, and pure-bred Arabians. It seems, therefore, that this formula for estimating the age of horses should not be used for all horses but instead should be adapted to different types or breeds. Due to the different origins of horse breeds, such as their selection, use, and criteria used for breed improvement, they could differ in the course of their development processes, which may be associated with the intensity of changes taking place on their teeth. Differences in the rate of teeth grinding between different breeds of horses may therefore have genetic and behavioral backgrounds, but it is also possible that external factors (nutrition and management) also play an important role. Based on literature sources, Muylle et al. [11] suggest that the cause of delayed tooth eruption may be malnutrition. In our study, it was found that Icelandic horses were in good condition. Slower abrasion of teeth of Icelandic horses might be a characteristic feature of primitive horses; given that from the beginning of their existence these horses were bred in purity, were maintained in conditions similar to the original natural environment, and were fed natural feeds, this feature probably perpetuated in their genotype. However, the nature of an animal’s diet may influence the abrasion of its incisors [11]. Hillson [23] stated that dental wear is not only caused by grinding of opposing crowns against one another, but also due to abrasive contact with food particles such as phytoliths, which form part of the structure of grasses. During the mastication process, the generated forces have an effect on the teeth and also on the periodontal ligament. It has been shown in humans that actual masticatory forces differ depending on the hardness of food [24]. It is expected that for horses the masticatory forces differ by type of feed [25], which may result in differences in rubbing of teeth, despite the general rules of dental assessment of age. Furthermore, incisors are generally unsupported by adjacent teeth and therefore might additionally undergo shearing forces during feed grinding [26]. Horses fed with high concentrate feeds might have dental irregularities associated with smaller mandibular excursions and might require special attention paid to the type of food consumed during dental examination [27]. Lowder and Mueller [28] suggest that veterinary surgeons, as well as horse breeders and practitioners, due to their work, should be able to determine the age of a horse based on the appearance of its teeth. Increasing interest in horses in recent years in the fields of recreation, amateur sport, agrotourism, and hippotherapy means that such a skill, at least to a limited extent, would be useful for many owners or users of horses. Accurately estimating a horse’s age can greatly facilitate solving problems related to the purchase and insurance of a horse, planning its future career, or performing therapeutic treatments.

Analysis of the usefulness of the examined features of incisors to determine the age of Icelandic horses in different age groups indicated that the least number of errors were made in the youngest horses (i.e., those up to two years of age, at a time when the eruption and growing of deciduous incisors and the disappearance of cups can be considered). Similar to Hucul horses [21], Thoroughbred [15], pure Arabians, trotters, and draft horses [17], the effectiveness of recognizing the age on the basis of specific features of incisors decreased significantly with the age of the Icelandic horse, reaching the highest percentage of errors in the group of 5–11-year-old horses. According to both Richardson et al. [14] and Nicks et al. [19], the younger the horses, the more likely it was to determine their age correctly based on the appearance of their teeth. The opposite tendency was observed in warmblood horses when estimating the age with the same tested method as in this work, in which the percentage of errors in the group of youngest horses was significantly more than twice as high as in the oldest horses [22]. The eruption of deciduous incisors was considered the best feature for reliable determination of age also in Hucul horses [21]. However, the percentage of errors made during this period was much higher (12%), and unlike Icelandic horses, it resulted in overestimating the age due to the earlier growing of deciduous intermediate and corner incisors. In Shetland ponies and donkeys, the eruption of the deciduous incisors followed later in comparison to the adopted scheme [11], therefore some people believe that the growth of permanent incisors is a more useful criterion for determining the age of horses [16,29]. According to Richardson et al. [30], the eruption of incisors is a permanent feature but not very reliable. Due to the specific construction of horse incisors, with a tooth funnel and the crown containing a tooth bay, called a cup, of a specific depth, it would seem that knowing the rate in which attrition occurs during the year [7] would enable more precise determination of the age of the horse based on these characteristics. Therefore, some authors suggest that at the age of 5–11 years, when the cups on permanent incisors disappear, estimating the age of a horse on this basis is the most precise method [10,14,19]. Others, however, due to breed and individual differences in relation to attrition of the incisors, enamel, and dentin hardness, as well as the depth of the cups and the position of corner incisors on dental arches, do not consider this to be an objective feature [11,16,20,31,32]. In a study of Hucul horses [21] the disappearance of cups on permanent incisors was not considered to be a useful criterion to estimate the age of primitive horses of a pony type. This is confirmed by the results of this work, which showed that when using this feature in Icelandic horses, errors occurred in as many as 68% of cases. Such a large discrepancy in comparison to the actual age of the horses may result from the specifics of the developmental processes for this horse breed. Łuszczyński et al. [10] showed a significantly higher percentage of errors when estimating the age of Arabian horses, which at the same time was significantly more often overestimated than underestimated compared to Anglo-Arabian horses. Changes occurring faster on the incisors of Arabian mares than Anglo-Arabian mares may indicate the achievement of an earlier degree of somatic maturity by horses of this breed. Studies confirming this thesis were carried out by Łuszczyński et al. [33] in research that analyzed the age of ossification of the metaphyseal growth plate and the growth rate of body dimensions. In our analysis of errors made while estimating the age of Icelandic horses based on the appearance of incisors, the significant differences were noted in two cases in which the most mistakes were made. In the group of horses aged 2 to 5 years, in which the exchange of deciduous incisors for the permanent incisors was taken into consideration, the age of the horses was more often underestimated than overestimated, which may indicate that in this period, the rate of growth and maturation of Icelandic horses was slow. In contrast, in 6–11-year-olds horses, their age was more often overestimated than underestimated, so the course of development processes likely increased in intensity. Strand et al. [34] claimed that the age of ossification of various appendicular growth plates of Icelandic horses was similar to horses of other breeds based on a radiological analysis. However, they believed that the rate of growth of this breed is slower and postulated the need for further research confirming this thesis. It can be assumed that a slower growth rate and longer time to reach the somatic maturity of primitive horses such as Icelandic horses, compared to warmblood horses, is the answer to the difficult living conditions in which these breeds were created. It is likely that these specific genetic features were supposed to protect against unfavorable factors of the external environment and provide natural protection against the formation of, among others, orthopedic developmental diseases. Such diseases, especially osteochondrosis, are rare in ponies or primitive horses [35] in contrast to warmblood horses [36], where they cause huge economic losses [37,38].

## 5. Conclusions

It can be stated that the estimated ages of Icelandic horses based on the physical appearance of their teeth did not match the horses’ real ages in more than a third of the cases. However, the percentage of mistakes made in relation to other breeds, mainly warmblood ones, was much smaller, which may be due to the origin and specificity of the Icelandic breed. The average percentage of errors made in the assessment of age based on the eruption and growth of deciduous incisors was significantly smaller compared to the estimated age based on the replacement of deciduous to permanent incisors, the disappearance of cups, or changes in the shape of the grinding surface. Significantly more frequent underestimation of age based on replacing deciduous for permanent incisors and significantly more frequent overestimation of age on the basis of the disappearance of the cup may indicate that Icelandic horses up to 5 years of age are characterized by a slower rate of growth than horses of other breeds, especially warmblood horses. Patterns used to determine the real age of horses based on changes occurring on their incisors should be modified in order to consider the specificity of the course of growth and maturation processes of horses of various types and breeds.

## Figures and Tables

**Table 1 animals-09-00298-t001:** A method for assessing the horse’s age based on the appearance of incisors [12].

Dental Criterion	Central Incisors	Intermediate Incisors	Corner Incisors
Eruption of: Deciduous Permanent	0–14 d 2, 5 y	4–6 wk 3, 5 y	6–9 mo 4, 5 y
Disappearance of the cups on deciduous incisors	1 y	1, 5 y	2, 0 y
Disappearance of the cups on permanent incisors Lower Upper	6 y 9 y	7 y 10 y	8 y 11 y
Grinding surface shape Round (lower/upper) Triangle (lower/upper) Biangle	12/15 y 18/21 y >24 y	13/16 y 19/22 y >24 y	14/17 y 20/23 y >24 y

d—days of age; wk—weeks of age; mo—months of age; y—years of age.

**Table 2 animals-09-00298-t002:** Percentage of errors in age assessment based on selected traits of incisors in comparison to the actual age of Icelandic horses.

Age Group (Years)	Dental Criterion	*n*	% Errors		
			Total	Overestimation of Age	Underestimation of Age
0–2.0	Eruption of deciduous incisors and disappearance of the cups on deciduous incisors	48	2.1 ^ABC^	-	2.1
>2.0–5	Eruption of permanent incisors	43	55.8 ^A^	13.9 ^a^	41.9 ^a^
>5–11	Disappearance of the cups on permanent incisors	25	68.0 ^B^	56.0 ^b^	12.0 ^b^
>11	Modifications of incisors shape	10	40.0 ^C^	20.0	20.0
Total		126	36.5	17.5	19.0

a, b—values with the same small letters differ significantly (*p* ≤ 0.05); A, B, C—values with the same capital letters differ significantly (*p* ≤ 0.01); Actual *p* value for: A (*p* < 0.001), B (*p* < 0.0001), C (*p* = 0.0012), a (*p* = 0.0425), b (*p* = 0.0257).

**Table 3 animals-09-00298-t003:** Differences between the estimated ages of Icelandic horses as determined on the basis of the appearance of their incisors and their real ages.

Age Group (Years)	*n*	Overestimation of Age			Underestimation of Age		
		$\bar{x}$ (Years)	Min (Years)	Max (Years)	$\bar{x}$ (Years)	Min (Years)	Max (Years)
0–2	48	-	-	-	1.0	1.0	1.0
>2–5	43	0.8	0.5	1.5	0.5	0.5	1.5
>5–11	25	1.0	0.5	3.5	1.0	1.0	1.5
>11	10	5.0	3.5	6.5	3.0	0.5	5.5
Total	126	1.3	0.5	6.5	0.9	0.5	5.5

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
