# Peer review of "The Frequency of Errors in Determining Age Based on Selected Features of the Incisors of Icelandic Horses"

_animals, 2019, doi:10.3390/ani9060298_

Round 1
Reviewer 1 Report
This is an interesting topic which should be of interest to horse practitioners and veterinary/clinical professionals world wide. The paper requires some tidying up in terms of phrasing, wording and some structural issues. These are highlighted in the attached file.

Author Response
Dear Reviewer,
thank you very much for the detailed review. We have corrected the manuscript (file attached) according to your suggestions. Please, take under consideration that in the manuscript there are also amendments for Reviewer 2.
We would like to explain that coldblooded horses are the subline of Døle Horse (citation from Bjørnstad, G.; Røed, K.H. Breed demarcation and potential for breed allocation of horses assessed by microsatellite markers. Anim. Genet. 2001, 32, 59–65.)
All your suggestions and comments helped us a lot and they improved the quality of our work.
Kind regards
Authors
Reviewer 2 Report
Review
The frequency of errors in determining age based on 2 selected features of the incisors of Icelandic horses
A nice simple study, well executed and generally clearly presented. I have made some suggestions that I feel would improve the manuscript further and support the reader less experienced in equine aging to understand the study for the authors to consider.
Simple summary
Line 15: please change ‘it’ suggest amending to ‘However, this method can play an auxiliary role in identifying age in horses of unknown origin’ to improve clarity
Line 18: suggest amending ‘The’ to ‘This study’
Line 20: I don’t think you need to include the start of the sentence, suggest removing ‘It can be stated that’ and just have ‘Determining the age of…’ to improve flow
Line 24: stray space between incisors and , - please remove
Abstract
Line 28: suggest replacing ‘it’ with ‘this’
Line 32: suggest replacing ‘The study’ to ‘This study’
Line 34: suggest rephrasing to ‘Age was determined on the inspection of the teeth and was compared …’
Line 38: it would be good to provide the reader with both the mean under / overestimation and the standard deviation here to show the variation in estimates recorded
Line 38: suggest replacing ‘considering’ with ‘within’ to improve synthesis
Line 47: I am not familiar with the use of noble breeds as common vernacular related to horse breeds, suggest replacing with different terminology which is more common across global audience of the journal (would this phrase relate to warmbloods and / or Thoroughbreds?)
Line 48: suggest starting sentence with ‘These results suggest that patterns…’
Keywords: age determining is a little odd – maybe change to ageing horses or horse age and scope to also include equine as a key word
Introduction
Line 54: century should be capitalised
Line 71: I wonder if there is scope to summarise some of the key morphological characteristics if equine teeth here as not all readers will be familiar with equine dentition
Line 74: suggest replacing ‘it’ with ‘which’ - I would also recommend outlining how age is determined here for a standard horse (in a simplified manner) or refer to an appropriate source for further information to support readers less familiar with this process / species. As you provide this in method, maybe summarise here and refer to table or move table 1 to here as better fit for introducing how to do this.
Line 83: amend ‘the study’ to ‘this study’
Method
Lines 97 to 99: amend to past tense
Line 100: please provide details of the extensive experience of the experimenter – qualifications etc
Line 111: insert ‘was’ before determined and insert ‘and’ between teeth and was
Line 114: suggest replacing ‘was used to show’ to ‘identified if significant differences …’
Line 115: please amend to ‘Analysis was undertaken using…’
Table 1: please include a legend and identify abbreviations at end of title or in legend to enable table to stand alone
Within your method, you should state how you determined the actual age of the horses in the sample (e.g. passport?) to demonstrate that these were accurate in the first place
Results
At the author’s discretion: given your sample was of mixed sex, it would be really interesting to also calculate and include results across mares and geldings independently as well as the across the total cohort for comparison.
Lines 125 to 127: please provide standard deviation as well as mean for over and underestimation of age
Table 2: I would like to see the actual P values for your results – could you integrate these into table 2 as an additional column perhaps
Tables 2 and 3: please include legends and outline of abbreviations used to enable these to stand alone
Within Table 3 could you also integrate the number of horses in each age group, suggest adding (n=xx) after info provided in age group (years) column
Line 136: advise replacing ‘considering’ with ‘within’
Line 165 and 166: remove commas
Results – did you take any photos of the horse’s teeth and characteristics? If you did it would be beneficial to the reader if you included some examples to provide visual cues which could support the text results and could illustrate the features you are describing
Discussion
at the moment your discussion is a little limited and focuses on your results in context of prior research but misses opportunities to consider some of the wider discussion points such as why these differences occur between breeds, should we be manually aging horses, what are the implications to under / over estimation of age for horse management, welfare etc - would suggest incorporating discussion of these into your manuscript
Line 178: include ‘was’ between horses and not
Line 186: please replace noble horses with more commonly used terminology
Lines 177 to 195: in this paragraph, I would consider adding in some discussion of why breed related dentition could occur, this could be linked to selective breeding for type and purpose then related to differences in management aligned with these
Line 206: replace noble as per earlier comments
Line 220: suggest replacing ‘they attrition’ with ‘ which attrition occurs’
Line 227: remove ‘an’ superfluous
Lines 196 to 254: this paragraph would benefit from discussion of why age related developmental differences may occur in Icelandic horses for those readers less familiar with this breed
Conclusion
Line 256: in conclusion is not required
Line 259: please replace noble horses with alternative terminology
Line 260: suggest changing ‘this’ to the Icelandic breed’
Line 263: remove extra space before comma
Author Response
Dear Reviewer,
thank you very much for the detailed review. We have corrected the manuscript (file attached) according to your suggestions. Please, take under consideration that in the manuscript there are also amendments for Reviewer 1.
We would like to explain that study taking under consideration additional factor (sex of studied horses) would be very interesting, however, due to the small size of the studied cohort, the introduction of an additional factor to statistical analyzes would result in a reduction in the size of groups hindering statistical analysis. The photos we took during the tests turned out to be technically very weak because the horses were constantly moving during the examination. Unfortunately, these photos are not suitable for publication.
All your suggestions and comments helped us a lot and they improved the quality of our work.
Kind regards
Authors
Round 2
Reviewer 2 Report
I would like to thank the authors for the clarity with which they have identified how they have revised the manuscript, which have addressed by feedback and I feel the additional sections have added to the discussion and will support readers less familiar with equine dentition to get the most from the paper. Subject to a couple of minor amendments below, I am happy to recommend publication - really interesting read.
Line 81: suggest change ‘on grass’ to ‘on natural forage’
Line 191: suggest influence not influences